

# You can hide but you can't run: apparent competition, predator responses and the decline of Arctic ground squirrels in boreal forests of the southwest Yukon

Jeffery R. Werner[1], Elizabeth A. Gillis[2], Rudy Boonstra[3] and Charles J. Krebs[4]

[1] Biodiversity Research Centre, Department of Zoology, University of British Columbia, Vancouver, British Columbia, Canada
[2] Department of Resource Management and Protection, Vancouver Island University, Nanaimo, British Columbia, Canada
[3] Department of Biological Sciences, University of Toronto, Scarborough, Toronto, Ontario, Canada
[4] Department of Zoology, University of British Columbia, Vancouver, British Columbia, Canada

Corresponding author
Jeffery R. Werner,
werner@zoology.ubc.ca

## ABSTRACT

Throughout much of North America's boreal forest, the cyclical fluctuations of snowshoe hare populations (*Lepus americanus*) may cause other herbivores to become entrained in similar cycles. Alternating apparent competition via prey switching followed by positive indirect effects are the mechanisms behind this interaction. Our purpose is to document a change in the role of indirect interactions between sympatric populations of hares and arctic ground squirrels (*Urocitellus parryii plesius*), and to emphasize the influence of predation for controlling ground squirrel numbers. We used mark-recapture to estimate the population densities of both species over a 25-year period that covered two snowshoe hare cycles. We analysed the strength of association between snowshoe hare and ground squirrel numbers, and the changes to the seasonal and annual population growth rates of ground squirrels over time. A hyperbolic curve best describes the per capita rate of increase of ground squirrels relative to their population size, with a single stable equilibrium and a lower critical threshold below which populations drift to extinction. The crossing of this unstable boundary resulted in the subsequent uncoupling of ground squirrel and hare populations following the decline phase of their cycles in 1998. The implications are that this sustained Type II predator response led to the local extinction of ground squirrels. When few individuals are left in a colony, arctic ground squirrels may also have exhibited an Allee effect caused by the disruption of social signalling of approaching predators.

## INTRODUCTION

The comprehensive role that snowshoe hares (*Lepus americanus*) play in the food web dynamics of North America's boreal forest epitomizes the notion of "foundation species" (sensu *Dayton, 1972*) who, by virtue of their abundance and influence on other species, help to define an entire ecological community (*Sinclair & Krebs, 2001*). In particular, the

cyclic oscillating abundance of the hare and its predators (e.g., lynx, coyotes, great horned owls) is central to our understanding of how predators influence the population dynamics of other common prey species (*Boutin et al., 1995*).

Predator-mediated "apparent competition" (*Holt, 1977*) between two prey can occur when at least one natural enemy is held in common and, that enemy enables the numbers of one species to negatively influence the abundance of another. In the case where such negative influences are not reciprocated between both prey species, 'asymmetric' apparent competition can lead to a variety of outcomes that depend on the intrinsic nature of the predator–prey relationship (*Sinclair & Pech, 1996*; *DeCesare et al., 2010*). Additionally, when predator–prey systems are routinely not at equilibrium, the primary prey population may achieve rapid increase leading to predator satiation and a temporary decrease in the per capita predation rate for both primary and secondary prey species (*Holt & Lawton, 1994*; *Abrams & Matsuda, 1996*). The effects of predator satiation, however, are limited to short time scales, because higher prey availability triggers a numerical response in the predator over subsequent predator generations (*Holt & Kotler, 1987*). Although the influence of such positive indirect effects is generally fleeting, this effect may recur when population densities of the primary prey show cycles that result in repeated satiation of the shared predators and reduced predation on the secondary prey species (*Abrams, Holt & Roth, 1998*). The periodic intensification and relaxation of predation is implicated in the synchronous population fluctuations of voles and either hares (*Angelstam, Lindström & Widen, 1984*) or grouse (*Hörnfeldt, Löfgren & Carlsson, 1986*) in Sweden, voles and shrews in northern Europe (*Hansson, 1984*; *Korpimäki et al., 2005*), and small rodents and ground nesting birds in northern Eurasia (*Sutherland, 1988*) and the high arctic (*Bêty et al., 2002*). The alternating influence of apparent competition during the decline phase of the cycle, followed by positive indirect effects leading to temporary escape from predator regulation during the increase phase, are sufficient conditions for synchronous cycling of primary and secondary prey (*Abrams, Holt & Roth, 1998*; *Norrdahl & Korpimäki, 2000*).

In North America, dramatic fluctuations in hare density are also known to entrain other prey species into cycles of similar duration (*Boutin et al., 1995*; *Krebs, Boutin & Boonstra, 2001*). The best documented case is that of the arctic ground squirrel (*Urocitellus parryii*; hereafter AGS), whose numbers in the SW Yukon varied in synchrony with hares for over three decades (*Werner et al., 2015a*). The putative mechanism for these coincident patterns in abundance is prey-switching, from hares to ground squirrels, during the decline phases of the hare cycle (*Boutin et al., 1995*; *Byrom et al., 2000*; *Krebs et al., 2014*).

These forest ground squirrel populations and their cyclic behaviour have been investigated since the 1970's (*Green, 1977*; *Hubbs & Boonstra, 1997*; *Byrom et al., 2000*; *Karels et al., 2000*; *Gillis et al., 2005*; *Donker & Krebs, 2012*). From 1973 to 1999, increases among AGS have been stopped by declines that recur with near decadal regularity. However, after 2000, AGS populations declined rapidly at lower elevations (∼900 m asl) in the Kluane region (*Donker & Krebs, 2011*), and colony extirpation is now widespread throughout similar habitats of the southern Yukon (colony occupancy = 4.2%; Table 1 in *Werner et al., 2015a*). The range of this historically ubiquitous herbivore appears to have contracted over the course of a decade. *Werner et al. (2015a)* hypothesized that this abrupt shift in

squirrel abundance, followed by a prolonged phase of very low numbers, was diagnostic of predator regulation (i.e., a 'predator pit' whereby a prey population is maintained at a lower stable equilibrium point well below carrying capacity). Nevertheless, no direct evidence to support this claim was offered at that time. Here we test whether the observed population dynamics of AGS are consistent with patterns predicted by predator–prey theory.

To predict whether this historically common herbivore (*Boonstra et al., 2001*) might regain a foothold in this system requires an explicit understanding of the predator response, especially as it relates to low prey density (*Sinclair & Krebs, 2002*). In cases where most prey mortality is caused by predation, the predator response can be determined by inspection of the instantaneous rates of change in the prey species over a realistic range of prey densities. Such analyses across a wide range of taxa (*Messier, 1994*; *Sinclair et al., 1998*) confirm two general categories of predator response curves predicted by predator–prey theory (*Holling, 1959*; *Holling, 1973*; *Sinclair & Pech, 1996*). Where predators suppress but do not eliminate prey at low density (Type III predator response; Fig. 1), persistence of the prey species is possible, as is the potential for population recovery in the absence of harmful stochastic events or strong fluctuations in food availability. Even where localised extirpations have occurred, habitat recolonization via immigration is possible. In contrast, when predators continue to consume prey at low prey availability (Type II predator response; Fig. 1) local extinction may be inevitable. Discriminating which of these two classes of predation are in operation is an important first step towards ascertaining the fate of AGS in the boreal forest ecosystem.

Our purpose is to document a change in the strength of indirect interaction between sympatric populations of hares and arctic ground squirrels and to clarify the recent influence of predation in controlling ground squirrel numbers during the later stages of their disappearance. Both of these aims are met in this paper by analyzing over two decades of long-term population census data collected for hares and AGS in the Kluane region of the SW Yukon. Strong correlation between hare and ground squirrel numbers across a wide range of hare densities is the expected outcome of asymmetric apparent competition (*Holt, 1977*). Furthermore, the population trajectory of the ground squirrels at various population densities provides useful information concerning the dynamics of the predation at low prey numbers (*Sinclair et al., 1998*). We evaluate the following hypotheses to account for observed changes in ground squirrel abundance over a 25-year period.

Based on a visual inspection of census data we hypothesized a Type III predator response (Fig. 1). This hypothesis maintains that AGS populations are regulated at low densities by predators but not regulated by predators at high density (*Sinclair & Pech, 1996*). This would explain the apparent shift in AGS abundance to a persistent lower population size after 2000 (i.e., a predator pit). Predictions are that the summer (May–August) per capita growth rate of the prey plotted over prey density will exhibit two positive equilibrium densities separated by a boundary threshold. The alternative hypothesis is that the predator response is Type II. AGS populations are unstable at low density (de-regulated by predation) but escape predation at high density. This hypothesis predicts the existence of a single upper stable population equilibrium and a single lower unstable boundary below which prey density declines towards extinction (Fig. 1).
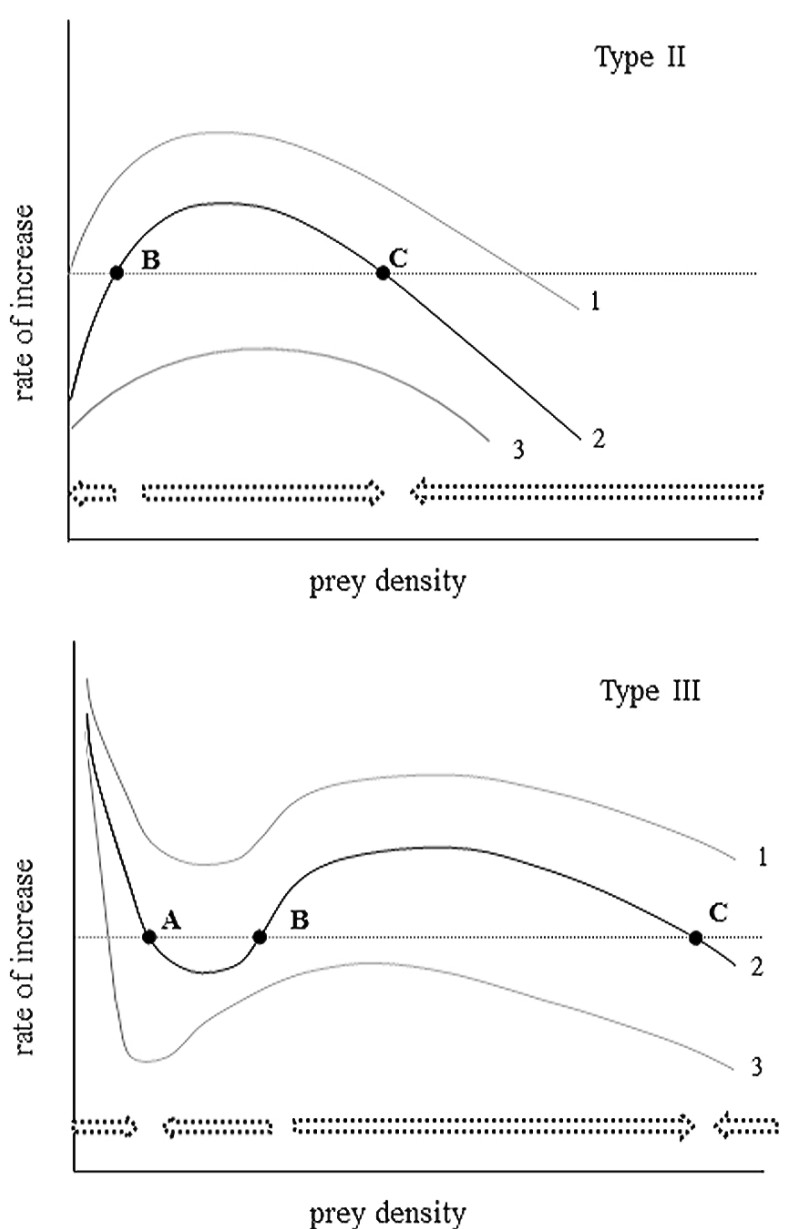

**Figure 1** **An illustration of the instantaneous rate of change for a prey population experiencing varying levels of Type II and Type III predation.** Arrows represent direction of projected population change relative to stable equilibria (A, C) and an unstable boundary threshold (B). Curves 1 and 3 represent different levels of predation rate (1 = lowest; 3 = highest). Key distinguishing features for a Type II is for declining prey population growth at low prey density, but for Type III a lower stable equilibria results from positive population growth at low numbers. Adapted from *Sinclair & Krebs (2002)*.

We additionally hypothesized that AGS populations were, at all times, governed by predator-mediated competition with hares followed by repeated escape from predator control. This hypothesis predicts a singular association between the annual census data of hares and ground squirrel throughout the monitoring period. A *priori* knowledge of the system indicates that this is quite likely (*Karels et al., 2000*; *Sinclair & Krebs, 2001*). On the

other hand, the correlation dynamic may also have weakened with time. The alternative hypothesis is therefore that a phase-change occurred after 2000, causing the system to shift from predator-mediated competition to a new state where squirrels are unaffected by hare numbers. This 'multi-state' alternative predicts an uncoupling in the strength of correlation between hare and squirrel density after 2000.

The distinguishing predictions for these hypotheses can be tested, given that two assumptions are correct: (1) changes to AGS numbers during the active season are not affected by overwinter mortality (as might be the case with annual census data only). (2) The bulk of AGS active season mortality can be attributed to predation. The first statement is axiomatic and the second supported by the literature. In the boreal forest 93–100% of active-season (summer) losses of radio telemetered AGS are confirmed predator mortalities (93% *Hubbs & Boonstra, 1997*; 96% *Byrom et al., 2000*; 100% *Donker & Krebs, 2012*; 100% *Werner et al., 2015a*).

## METHODS

### Study species

The arctic ground squirrel is a burrowing, semi-fossorial polygynous sciurid (*McLean, 1981*) inhabiting arctic and subarctic regions of North America and Russia (*Naughton, 2012*). They are typically found in open meadows and tundra, but one of the unique features of their geographic distribution is the extension of their range into the boreal forests of the Yukon and NW British Columbia. Squirrels hibernate for 7–9 months (September/October through April); the remaining active-season is short and individuals must reproduce and gain substantial mass (energy reserve) during this time (*Buck & Barnes, 1999*). Juvenile squirrels are born in May but remain in the natal burrow for nearly one month before emerging (with females being philopatric and males dispersing) to establish a territory, and achieve sufficient mass to survive winter hibernation (*Carl, 1971*; *Lacey, 1991*; *Sheriff et al., 2013*).

In northern ecosystems this common small mammal functions as an important prey item (*Hubbs & Boonstra, 1997*), herbivore (*Boonstra et al., 2001*) and as an ecological engineer (*Price, 1971*). Their distribution drives the abundance and spatial arrangement of other ecosystem constituents from carnivores to plant communities (*Wheeler & Hik, 2012*). In the SW Yukon ground squirrels are an important alternate food source for mammalian and avian predators such as the coyote (*Canis latrans*), lynx (*Lynx lynx*), Northern goshawk (*Accipiter gentilis*) and Red-tailed hawk (*Buteo jamaicensis*) (*Doyle & Smith, 1994*; *O'Donoghue et al., 1998a*; *O'Donoghue et al., 1998b*).

### Study area

The area is in the rain shadow of the St. Elias Mountains and receives a mean annual precipitation of ca. 230 mm, mostly falling as rain during the summer months, but including an average annual snowfall of about 100 cm (*Krebs, Boutin & Boonstra, 2001*). Squirrel and hare data were collected on two 9-Ha trapping grids, located several kilometers apart (~900 m above sea level (asl)) (61°00′38″N, 138°11′31″W and 60°55′53″N, 137°58′25″W). These grids were dominated by white spruce forest (*Picea glauca*), willow (*Salix* spp.) or bog birch (*Betula glandulosa*) thickets, and occasional aspen stands (*Populus tremuloides*)

(see *Boonstra et al., 2001* for detailed trap locations and *Turkington et al., 1998*; *Turkington et al., 2002* for regional descriptions).

## Population estimates

Squirrels and hares were trapped twice yearly (May and August) at two sites (GPC and Chitty grids; *Boonstra et al., 2001*) for 25 consecutive years (nearly 3 hare cycles; 1990–2015). The methods of data collection are fully described in *Boonstra et al. (2001)*. Population density estimates were obtained by mark–recapture methods (described below). Each mark–recapture session consisted of between two (typically) or four (in situations of low AGS/hare density) consecutive days of trapping in May/August for AGS and April/October for hares.

   Squirrels traps were set at 0800 h, checked every 1.5 h, and closed by 1230 h each trapping day. Arctic ground squirrels are highly trappable and recapture rates were high (>80%). Squirrels were live-trapped on two grids each with 50 traps spaced 30 m apart in a 10 × 10 pattern with traps placed at alternate grid stations. Live traps (14 cm × 14 cm × 40 cm; Tomahawk Live Trap Co., Tomahawk, Wisconsin, USA) were baited with peanut butter. Upon first capture, squirrels were transferred to a mesh bag, where they were then tagged in both ears with unique monel No. 1,005–1 tags (National Band and Tag Co., Newport, Kentucky, USA), weighed (Pesola spring scale ±5g), sexed, and measured for structural size (zygomatic arch width) using a 150 mm metric dial reading caliper.

   Snowshoe hare live-traps were pre-baited with alfalfa cubes for 3–5 days prior to being set. Trapping sessions consisted of 2–3 nights of trapping within a 5-day period in spring (early April) and autumn (October). The traps were set at 2,200 h and checked at 0600 h to minimise the length of time hares were detained. Upon capture, each hare was identified to sex, weighed with a Pesola spring scale (±10 g), its right hind foot length was measured (as an index of body size), and its right ear was tagged (No. 3 Monel tags; National Band and Tag Co., Newport, Kentucky, USA). Trapping and handling protocols were approved by the University of British Columbia Animal Care Committee in accordance with the guidelines of the Canadian Council on Animal Care (NIAUT certificate # 5740 –13) and all research was sanctioned under the Yukon Scientist and Explorers Research permit (License # 14-10S&E, file # 6800-20-43).

## Analysis

Population estimates and standard errors were computed using a mark-recapture heterogeneity (jackknife) model (*Pollock et al., 1990*) from Program Capture (*White et al., 1982*; *Rexstad & Burnham, 1991*). We calculated the effective sample area (to produce a density estimate) for each trapping period by adding a boundary strip to the edges of the trapping grid equal to half the mean maximum distance moved (*Otis et al., 1978*; *Krebs et al., 2011*). This method performs equally well to other density estimates for small mammals (*Krebs et al., 2011*). Rates of population increase were calculated for seasonal and annual time intervals. Yearly rates based on spring census data were used to generalise annual patterns of change over time. To distinguish between possible Type II and Type III predator relationships, the instantaneous rates of increase were calculated for each active-season (May–August). The shape of the predator prey relationship was determined by plotting the instantaneous per capita rate of change for AGS between $N_t$ and $N_{t+1} (dN/dT/Nt)$ over

population density at $N_t$ (*Sinclair et al., 1998*). Data from 1990 and 1999 were excluded from this plot because these years coincide with intense prey-switch events that result in total population collapse (such perturbation events are treated separately when analysing cyclic dynamics; *Sinclair & Krebs, 2002*). Because the application of goodness-of-fit curves to time series data is inappropriate, regression curves are used only to predict the regions where rates of change are zero. The slope of the regression curve where it crosses the zero line was used to classify stable equilibria and/or unstable boundaries.

All other statistical analyses were calculated using the programs JMP version 4.0 (SAS institute Inc., Cary, North Carolina, USA) or StatistiXL version 1.8. We measured correlation between hare and AGS densities for the time periods 1990–2011 and 2012–2014 by calculating the coefficient of determination using ordinary least squares regression. We used the Chow Test (*Chow, 1960*) to test if the coefficients of linear regression for each time period were equal. This is used in the analysis of time series data to test for structural breaks in the correlation dynamic of a single explanatory variable (*Chow, 1960*). Mann–Whitney $U$-tests were used to compare median population densities where data were not distributed normally.

## RESULTS

Both hare and ground squirrel population density were marked by repeated fluctuations (Fig. 2) that were coincident between 1990–2001 ($R^2 = 0.69$; Fig. 3) but not coincident between 2002–2013 ($R^2 = 0.01$; Fig. 3). The amount of variation in squirrel density explained by hare density ($R^2$ value) differed significantly between the two time periods (Chow Test: $F_{2,14}=33.4$, $p < 0.001$; Fig. 3). After 2000, AGS populations failed to increase despite a modest rise in hare numbers including peak hare abundance in 2005 and again in 2015. Mean density for the period preceding the year 2000 was significantly larger (by one order of magnitude) than for the interval proceeding the pivotal population decline ($U = 41$, $n1 = 24$, $n2 = 26$, $p < 0.01$). Figure 4 illustrates two patterns: first, the annual rate of change for the decline phase of the hare cycle follows a different track from that for the increase (circular pattern to the right); and second, the overall dynamic changed from visually circular to unstable below a threshold density of 0.5 AGS individuals/ha (Fig. 4).

To estimate the predator response, the instantaneous rates of per capita population growth for the active-season period (May–August) were plotted against spring population density. Over the entire range of densities recorded for AGS the density dependence relationship is curvilinear (Fig. 5), and conforms to a Type II predator–prey relationship (Fig. 1). The rate of population change crosses the zero line at two specific density locations. The upper stable point in Fig. 5 is equivalent to point C in Fig. 1; here density dependent processes act to maintain squirrel density at $\sim$2/Ha, which is the historical carrying capacity (*Werner et al., 2015a*). The negative downward slope of the population function at point C is indicative of population regulation (*Sibly & Hone, 2002*). In contrast, the lower threshold ($\sim$0.25–0.5/Ha) is an unstable boundary below which AGS abundance can decline toward extirpation (Fig. 5). Because density dependence varies from values which are weakly positive to negative, there is little compensation for stochastic effects at or near this threshold.

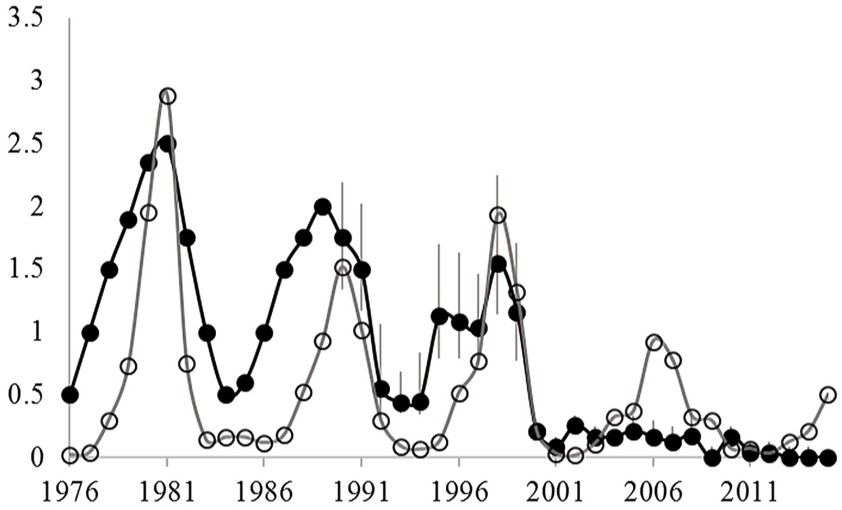

**Figure 2 Changes in the spring density of arctic ground squirrels in two live-trapping grids in the boreal forest at Kluane Lake since 1977 (black circles) and the spring density of snowshoe hares in the same habitat (grey circles).** Data from *Werner et al. (2015a)*; AGS density estimates for 1977–1989 are based on an index of abundance, while 1990–2015 are mark-recapture estimates. Grey vertical bars are 95% confidence limits.

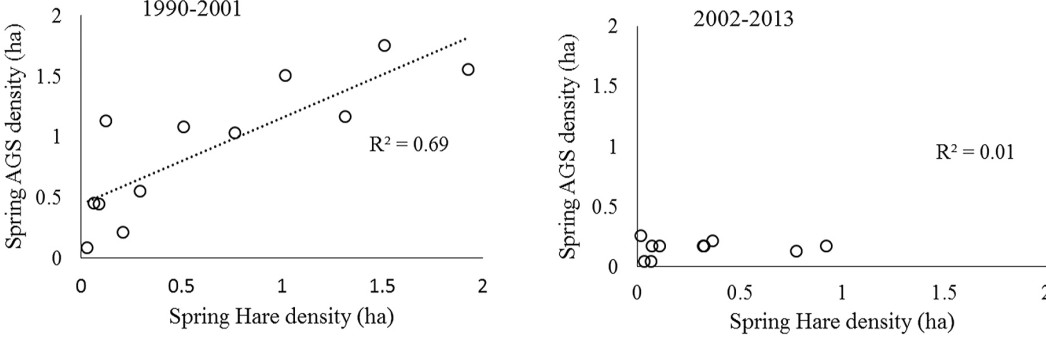

**Figure 3 The historical relationship between spring density of sympatric populations of snowshoe hare and arctic ground squirrels in the boreal forests of the Kluane region, SW Yukon.** The time periods represent conditions before and after the population collapse of ground squirrels. A high correlation is consistent with apparent competition between the two species mediated by shared predators.

## DISCUSSION

### Study limitations

Our aim is to infer the nature of the predator–prey relationship and to assess prospects for population recovery. A variety of *a priori* reasons exist to expect predation to show strong signals through the noise of contingent events in this part of the boreal forest (*Krebs, Boutin & Boonstra, 2001*; *Donker & Krebs, 2012*; *Werner et al., 2015b*). However, the possible contribution of Allee Effects (*Allee, 1931*) to the extinction process must first be considered. Social processes can cause per capita rates of change similar to those seen in Type II predator interactions, but without recourse to predation (*Odum & Allee, 1954*).

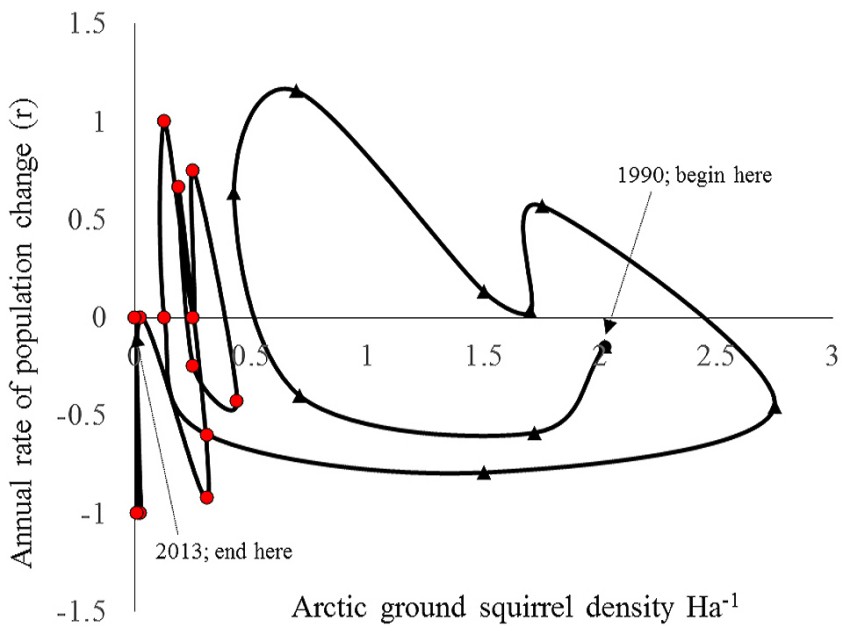

**Figure 4** **Rate of increase (*r*) for arctic ground squirrels in relation to its population density between 1990–2013 (annual spring trapping records; Kluane, SW Yukon).** The data comprise one cycle and subsequent non-cyclic dynamics post 2000. The rate of AGS increase follows a different path from that of the increase phase due to apparent competition with the snowshoe hare (first 10 data points). The rates exhibit unstable dynamics below a density threshold of 0.5/ha that end in eventual localised extinction (final 13 data points).

The range of mechanisms that could contribute to Allee Effects for AGS include lower mate finding success (*Stephens & Sutherl, 1999*), lowered reproduction resulting from poor body condition, reduced group vigilance and alarm calling (*Stephens & Sutherl, 1999*), the loss of conspecific cues for habitat choice (*Reed & Dobson, 1993*), and other forms of behavioural dysfunction (*Brashares, Werner & Sinclair, 2010*).

Because the mating system of the arctic ground squirrel is polygynous, mate-finding limitation caused by skewed sex ratios at low population size is not likely to cause inverse density dependence. It has been shown that adult females from this population over the same time period were in good hibernating condition by autumn (*Werner et al., 2015b*) and, because these females hibernate singly, their reproductive output the following spring cannot be attributed to any socially-mediated or density related process prejudicial to reproduction. Conspecific attraction may play a role in low rates of recolonization, but this process will not hasten population decline because resident females are philopatric (*Carl, 1971*) and site abandonment has never been observed. Even so, the lost benefits of predator detection and signalling behaviour common to ground squirrels (*Sherman, 1977*) and other social rodents (*Blumstein, 2007*) remains a possible contributing factor. Be it noted that although Allee effects may aggravate population declines and/or constrain population recovery, they are inadequate to trigger such a decline (i.e., operates at low density).

It has been shown theoretically (*Holt, 1977*) and empirically (*Pech, Sinclair & Newsome, 1995*) that when the abundance of primary prey increases, the attendant increase in

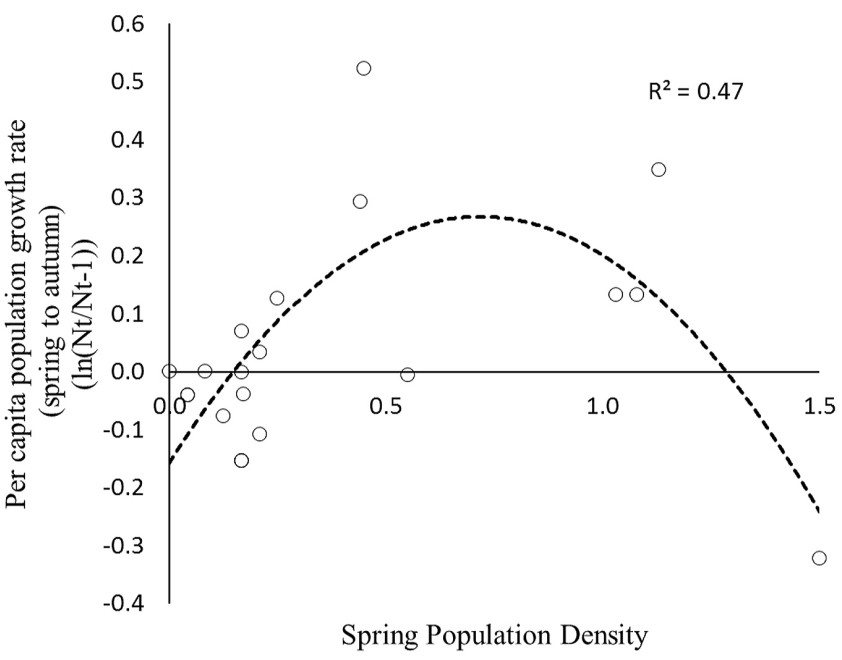

**Figure 5** **The instantaneous rate of population increase for the arctic ground squirrel during the non-hibernation period (May–August) plotted against spring population density 1990–2013.** This hump-shaped relationship is indicative of type II predation given that nearly all mortality during these months is due to predation.

predators results in a decline in the abundance of secondary prey. But, when the preferred prey undergoes dramatic cycling, shifting predator preferences at different phases in this cycle can result in similar fluctuations and coincident declines in the abundance of both prey (*Norrdahl & Korpimäki, 2000*). The strongly asymmetrical nature of indirect effects purported here (changes in hare abundance cause similar changes in ground squirrels but not vice versa) precludes using AGS extirpations as a natural test of these indirect effects. Only the removal of hares could serve as such a test. Despite this, the large and nearly coincident fluctuations in both hare and ground squirrel numbers (Fig. 2), coupled with the observation that ground squirrels do not cycle in the absence of hares (*Donker & Krebs, 2011*), provide compelling evidence for the existence of these indirect interactions up until the collapse of 2000. Further evidence for the role of predator mediated indirect effects for synchronising population oscillations of these sympatric prey species come from (i) detailed studies linking AGS cycles to intensified predation during their decline phase (*Hubbs & Boonstra, 1997*; *Byrom et al., 2000*), and (ii) the observation that the density, hunting behaviour, and diet of these same predators alter predictably with hare abundance (*O'Donoghue et al., 1997*; *O'Donoghue et al., 1998a*; *O'Donoghue et al., 1998b*).

Depensatory predation is possible in multi-prey systems where predators persist irrespective of secondary prey abundance (*Holt & Lawton, 1994*). Where a curvilinear (hump-shaped) density response exists, stochastic prey population losses are not compensated for by positive density dependent processes at low population size. In such cases of inverse density dependence, populations below a certain threshold will trend to extinction

(*Sinclair & Krebs, 2002*). Here we demonstrate a type II predator–prey relationship and quantify the critical density threshold (>0.7/ha) necessary for the persistence of AGS in the Yukon boreal forest system where hares are the preferred prey species for a range of predators (*O'Donoghue et al., 1998a*). The breach of this lower boundary during 1999–2000 is the most likely explanation for the loss of alternating apparent competition and indirect mutualism/commensalism with the snowshoe hare, and for subsequent loss of cyclicity in AGS abundance. These findings indicate that, as a secondary prey species, AGS may succumb to depensatory mortality from predator populations that are otherwise sustained by an abundant primary prey (*DeCesare et al., 2010*). The existence of prey switching during specific periods of the hare cycle further exacerbates this unstable dynamic by reducing the length of time squirrels remain at or near carrying capacity.

Upon first inspection, the decade long persistence of ground squirrel numbers at very low densities appears to corroborate a lower stable equilibrium characteristic of a type III predator interaction (Fig. 2). Inspections of this yearly time series prompted *Werner et al. (2015a)* to hypothesize the possible existence of a predator pit at low numbers. However, when changes to population density are limited to marked individuals caught at the beginning and end of each active-season we find that local populations sometimes dropped to zero in autumn, only to exhibit positive numbers the following spring. The existence of multiple extirpation events on the same trapping grids were effectively obscured in yearly census data because unmarked dispersing immigrants settled into the newly vacant habitat (*Donker & Krebs, 2012*). This discrepancy underscores the important fact that the resolution of population census data (annual vs. seasonal) must be fitted to the needs imposed by the research question. In this case it was important to estimate the predator response by restricting our analysis to rates of change for the active-season only.

Because density dependent relationships are notorious for their non-repeatable characteristics over time and space (*Krebs, 2002*) the patterns we report may have limited wider application. Yet, given widespread concomitant disappearance of AGS from boreal forests in Kluane (*Gillis et al., 2005*; *Donker & Krebs, 2011*) and other low elevation habitats of the southern Yukon (*Werner et al., 2015b*), the intensification of predation across large areas cannot be discounted as the proximate cause of these patterns. The most recent surveys of Lynx abundance indicate that during the last low phase of the hare cycle (2008–2012) Lynx were more numerous than ever recorded for any previous low (*Krebs et al., 2016*). In Kluane, Lynx focus on alternative prey like red squirrels (*O'Donoghue et al., 1998a*) during the winter months in years when hares are sparse. Higher predator abundance in advance of the increase phase of the hare cycle makes future indirect mutualism less likely because of the shortened lag in the predators' numerical response.

We propose that predation, being the most consistent explanation for population collapses, is the most likely proximate cause of local ground squirrel extinctions in the boreal forests of the SW Yukon. The principle assertions of this paper—the existence of Type II predation and an unstable critical threshold in prey density—is being tested by raising the local density of AGS above 0.7/ha in a series of experimental reintroductions into formerly occupied habitats within the boreal forest zone (*Werner, 2015*). This study offers a practical example of how monitoring the per-capita rate of change for prey species

can be used to infer the predator relationship and, by extension, the range of prey densities where mortality may be depensatory.

### Funding

Funding was provided from the Natural Science and Engineering Research Council of Canada, the Yukon Fish and Wildlife Enhancement Trust, the Northern Science Training Program of Environment Canada (administered by the University of British Columbia), The W. Garfield Weston Foundation Fellowship Program (a program of the Wildlife Conservation Society Canada funded by The W. Garfield Weston Foundation), the Northern Research Endowment fund (administered by Yukon College), a grant-in-aid from the Arctic Institute of North America and logistical support from Yukon Territorial government. The funders had no role in study design, data collection and analysis, decision to publish, or preparation of the manuscript.

### Grant Disclosures

The following grant information was disclosed by the authors:
Natural Science and Engineering Research Council of Canada.
Yukon Fish and Wildlife Enhancement Trust.
Northern Science Training Program of Environment Canada.
The W. Garfield Weston Foundation Fellowship Program.
Northern Research Endowment fund.

### Competing Interests

The authors declare there are no competing interests.

### Author Contributions

- Jeffery R. Werner conceived and designed the experiments, analyzed the data, contributed reagents/materials/analysis tools, wrote the paper, prepared figures and/or tables.
- Elizabeth A. Gillis and Rudy Boonstra performed the experiments, reviewed drafts of the paper.
- Charles J. Krebs conceived and designed the experiments, performed the experiments, contributed reagents/materials/analysis tools, reviewed drafts of the paper.

### Animal Ethics

The following information was supplied relating to ethical approvals (i.e., approving body and any reference numbers):

This research was approved by the University of British.

Columbia Animal Care Committee in accordance with the guidelines of the Canadian Council on Animal Care.

Completed the ethics training requirements of the Canadian Council on Animal Care (CCAC) / National Institutional Animal User Training (NIAUT) Program Certificate #: 5740-13.

### Field Study Permissions

The following information was supplied relating to field study approvals (i.e., approving body and any reference numbers):

BC Parks Permit 106618

Yukon Science & Explorers Permit License No 14-10 S&E.

### Data Availability

All data used in this paper are archived and freely available at the website http://www.zoology.ubc.ca/~krebs/kluane.html, and can be downloaded in an Excel file by clicking on "Kluane Monitoring Data."

### Supplemental Information

Supplemental information for this article can be found online at http://dx.doi.org/10.7717/peerj.2303#supplemental-information.

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
