# Peer review of "You can hide but you can’t run: apparent competition, predator responses and the decline of Arctic ground squirrels in boreal forests of the southwest Yukon"

_PeerJ, doi:10.7717/peerj.2303_

## Round 0.1 · original submission · Minor Revisions

Both reviewers had positive comments on the manuscript, and I don't have any reservations about it's ultimate acceptability. There are some minor points identified that you may wish to address, but the more substantive issues are those from reviewer 2. This reviewer questions whether or not the data are sufficient to justify characterization of the interactions as apparent mutualism. From these comments I understand that a more definitive understanding of the mechanism associated with synchrony requires exclusion experiments. Would you please consider these comments, especially the argument that you should narrow the focus regarding mutualism. If you disagree with this reviewer, I think it is important for you to anticipate this point from readers in revising your text, as well as in your rebuttal argumentation.

These points notwithstanding, I enjoyed your paper and I think it will offer a great example of predator-prey interactions.

·

Basic reporting

Manuscript: RI-2016:04:10438:0:1:REVIEW
Title: You can hide but you can’t run: apparent competition, predator responses and the decline of Arctic ground squirrels in the boreal forest of the southwest Yukon
Authors: Werner, Gillis, Boonstra, and Krebs
Reviewer: Gary J. Torrisi
Basic Reporting:
This paper looks at the reason(s) behind population collapse of a sympatric population of hares and Arctic ground squirrels. The authors propose that predation is the proximate cause of ground squirrel extinctions.
This paper offers an ample introduction supplemented with appropriate background information and supportive references. Several biological concepts were introduced and often explained for the reader to follow. I would suggest that “predator pit” be explained briefly for clarity.
This paper meets the criteria for “stand alone” reporting and contributes to the knowledge in this field of study.
A relatively clean manuscript with little in the way of rewrite is needed.
I suggest:
In the Introduction, place a period at the end of first paragraph on page 7.
In the Results section, rewrite the last paragraph sentence that begins with “Be it noted…”
In the Reference pages, there are 10 citations in the body of the manuscript with no reference listed.
There are two references in the list but I was unable to locate citations in the body of the manuscript (Efford, et al., 2009; Rexstad and Burnham, 1991).

Experimental design

Experimental Design:
Twenty five years of work has been expressed in this paper. Certainly reproducible by another researcher, however, I don’t have 25 years to do so.
Sound methods employed.
The authors mention “hare cycle” and I would like to see it expressed in terms of a reasonable time frame relative to the 25 year overall period of research time.
Methods, page 9, spell out KY to Kentucky to keep it uniform throughout the paper.

Validity of the findings

Validity of the Findings:
It is not clear to me if Peer J or the authors were to have supplied the raw data for review. However, I would have liked to have seen the results from GPC and Chitty side by side over the years. It brings to question, would it have been appropriate to have moved the trapping and recapture around the area tested? Would this have influenced your results? Just thinking out loud.
Final thought: would it be correct to surmise a Type III influence followed by a Type II collapse years later where the Type III effect ultimately led to a Type II extinction event?
Overall – well written

Additional comments

As I read your work over and over I couldn't help but think that this paper would serve as a classroom lesson introducing a number of biological concepts for discussion and study. A wonderful teaching tool for new biology students with limited backgrounds and knowledge.
Gary

Reviewer 2 ·

Basic reporting

This is a very interesting paper documenting some unusual changes in dynamics of two synchronously cycling rodents that share a predator. This is one of the classic datasets in Ecology, and it's really exciting to see it continue. The article generally conforms to PeerJ standards, although there are a number of specific clarity or technical problems (see below).

Experimental design

The primary criticism I have is with setting this up as a test or evidence of apparent mutualism. Although those could be the reason for synchronous fluctuations, the definitions of those terms require exclusion experiments to identify the type of mechanism. Simply observing a correlation between the two species over time is not a test of apparent mutualism.

Validity of the findings

The conclusion that the positive correlation between AGS and hares over time indicates apparent competition/mutualism is flawed. Other conclusions are fine, but see below for concerns about mismatched figures and text.

Additional comments

Fixing this just requires dialing back the rhetoric on apparent competition/mutualism throughout the MS -- the results are all the same. The primary points of the paper are that a) something happened in 2000, and b) the shape of the per capita growth rate vs. density curve for AGS is humped. That creates a predator pit, and could be the reason for the failure of AGS to bounce back after 2000. That's a nice story.


-- lines 71 through 75 A nice review of the apparent competition/mutualism literature, but this conclusion is too strong.
When populations cycle the shared prey do not necessarily cycle synchronously -- they *can* but they don't have to. Abrams et al define the indirect mutualism/competition as the response of the time-averaged density of species 1 when species 2 is removed. No focus on synchronicity at all.
-- line 112 -- just one hypothesis please.
-- line 115 through 124 -- just rewrite this as part of the next hypothesis, that predation is holding them down post 2000. The loss of synchronicity can be interpreted as moving to a new stable regime, but doesn't say anything about the mechanisms that created the synchronization in the first place.
-- Figure 2 -- should be SE on the points that have them (1990 - 2014 for AGS according to methods).
-- line 228 -- R^2 reported in text and in excel sheets doesn't match that in the figure.
-- line 238 through 249 -- The lower threshold reported in the text of about 0.5 /ha is nearly double that indicated in the
figure closer to 0.25. Also there are 16 points in the figure but only 14 years? R^2 in figure doesn't match figure in
excel sheet. In addition, the figure has AGS densities above 0.5, but if you look at figures 2 or 3 AGS are never above 0.5
after 1999. I'm not sure what's actually plotted in that figure.

---

## Round 0.2 · accepted · Accept

I very much appreciate your revisions and the detailed comments you offered in your rebuttal letter. I suspect I'll be using this paper in my ecology teaching!

·

Basic reporting

Upon completion of the second review process, I find that the the authors have made the necessary corrections and additions that have strengthened this manuscript. Notably, the elimination of indirect mutualism and the addition of clarification in the introduction and discussion sections. Furthermore, citations in the body of the manuscript and additional references were noted.

Line 388 (reference section) please add the year.
Line 123 - We... through line 125 requires rewrite

Experimental design

No Comments

Validity of the findings

No Comments

Additional comments

I found this investigation to be interesting and recognize areas of potential research related to this work that may add to the understanding of proximate cause and possible stochastic influence upon populations including Type II and Type III results.